# Antibacterial Gelatin Composite Hydrogels Comprised of In Situ Formed Zinc Oxide Nanoparticles

**DOI:** 10.3390/polym15193978

**Published:** 2023-10-03

**Authors:** Ya-Chu Yu, Ming-Hsien Hu, Hui-Zhong Zhuang, Thi Ha My Phan, Yi-Sheng Jiang, Jeng-Shiung Jan

**Affiliations:** 1Department of Chemical Engineering, National Cheng Kung University, Tainan 701, Taiwan; julie.yu.90427@gmail.com (Y.-C.Y.); e34064161@gs.ncku.edu.tw (H.-Z.Z.); n36087113@gs.ncku.edu.tw (T.H.M.P.); js3794490@gmail.com (Y.-S.J.); 2Department of Post-Baccalaureate Medicine, College of Medicine, National Chung Hsing University, Taichung 402, Taiwan; minghsienhu@gmail.com; 3Orthopedic Department, Showchwan Memorial Hospital, Changhua 500, Taiwan

**Keywords:** composite hydrogel, antimicrobial, gelatin, nanoparticle, in situ formation

## Abstract

We report the feasibility of using gelatin hydrogel networks as the host for the in situ, environmentally friendly formation of well-dispersed zinc oxide nanoparticles (ZnONPs) and the evaluation of the antibacterial activity of the as-prepared composite hydrogels. The resulting composite hydrogels displayed remarkable biocompatibility and antibacterial activity as compared to those in previous studies, primarily attributed to the uniform distribution of the ZnONPs with sizes smaller than 15 nm within the hydrogel network. In addition, the composite hydrogels exhibited better thermal stability and mechanical properties as well as lower swelling ratios compared to the unloaded counterpart, which could be attributed to the non-covalent interactions between the in situ formed ZnONPs and polypeptide chains. The presence of ZnONPs contributed to the disruption of bacterial cell membranes, the alteration of DNA molecules, and the subsequent release of reactive oxygen species within the bacterial cells. This chain of events culminated in bacterial cell lysis and DNA fragmentation. This research underscores the potential benefits of incorporating antibacterial agents into hydrogels and highlights the significance of preparing antimicrobial agents within gel networks.

## 1. Introduction

In recent years, the extensive use of antibiotics has caused the emergence of multidrug-resistant bacteria and a new crisis in biomedical fields. This not only has increased the risk and severity of infections in hospitals but also poses threats to human survival. Therefore, it is necessary to search for new methods to combat diseases caused by bacteria [1]. Nanotechnology has emerged as a highly potent and versatile tool in combating the challenges associated with multidrug-resistant pathogenic bacteria. Unlike antibiotics, nanoparticles pose a formidable hurdle for bacteria to develop resistance [2]. Among the inorganic particles, zinc oxide nanoparticles (ZnONPs) exhibit significant antibacterial properties and good biocompatibility with human cells [3], which makes them an attractive alternative for treating multidrug-resistant bacteria [4,5]. Many studies reported that adding inorganic zinc oxide or silver, copper, and gold NPs can provide composite hydrogel materials with effective antibacterial properties [1,4,6]. Hydrogels exhibit stable chemical and physical properties, good biocompatibility, and biological degradability, and allow for cell attachment and payload delivery [7,8]. Due to their excellent biological properties, hydrogels have the potential to be utilized in a wide range of biomedical fields [4,6,9,10,11,12]. ZnO nanocomposite hydrogels are typically synthesized through the doping of ZnONPs. Compared to composite hydrogels prepared by incorporating inorganic NPs into hydrogels [13,14], in situ formation is a better method that can avoid harsh reaction conditions such as high temperature, high pressure, an inert atmosphere, and the use of toxic solvents [15]. In situ mineralization of ZnONPs in hydrogel networks can be conducted by using gelatin which is derived from the partial hydrolysis of collagen and possesses a limited number of functional groups, including amino and carboxyl groups. These functional groups have the ability to chelate ions, serving as nucleation sites and controlling the size of in situ synthesized inorganic NPs. Gelatin is also a biocompatible, biodegradable, and non-cytotoxic polypeptide, which renders it an ideal candidate for the preparation of NPs in hydrogel networks [16]. Previous studies demonstrated the efficacy of gelatin for the mineralization process of gold (Au) or silver (Ag) NPs [17,18]. As a result, recent studies focused on the in situ formation of zinc oxide with different materials [19,20,21]. This process results in a uniform distribution of zinc oxide and excellent antibacterial effects, making the incorporation of ZnONPs within hydrogels a promising method to obtain antibacterial material for various biomedical applications. The antibacterial properties of ZnONPs rely on multiple bactericidal mechanisms. Reactive oxygen species produced by ZnONPs can disrupt bacterial cell membranes and lead to bacterial death [22]. Additionally, Zn^2+^ can inhibit bacterial metabolic processes, such as glycolysis and reaction involving phosphoenolpyruvate and F-ATPase, which can be bacteriostatic and enhance the bactericidal actions of other agents [23].

In this study, we report the synthesis of an antibacterial material obtained by grafting a methacryloyl substitution group from methacrylic anhydride onto gelatin, followed by the in situ loading of ZnONPs in the gelatin hydrogel. The functional groups in gelatin, such as amino and carboxyl groups, can chelate ions and provide nucleation sites for the formation of in situ ZnO nanoparticles. This functionalization of gelatin can also help to control the size of the formed ZnO nanoparticles. In addition, gelatin methacrylate (GelMA) polypeptides contain arginyl-glycyl-aspartic acid (RGD), a tripeptide that can promote cellular activities, as well as matrix metalloproteinase (MMP) sequences that induce tissue restoration [24]. Thus, this material is excellent for wound healing. We evaluated the physical properties of the gelatin hydrogel and ZnO-loaded gelatin hydrogels and the antibacterial activity of materials with different concentrations of ZnO.

### 1.1. Synthesis and Characterizations of GelMA Polypeptides

The method of synthesizing GelMA polypeptides was reported previously [25]. Gelatin (10 g, type A, Sigma-Aldrich, St. Louis, MO, USA) was dissolved in 100 mL of phosphate-buffered saline (PBS, pH 7.4, 0.2 N) to create a 10 wt% gelatin solution at 50 °C. Methacrylic anhydride (30 mL, MA, Acros Organics, Verona, Italy) was added and reacted for three hours. The solution was then dialyzed using a cellulose membrane tube (MWCO 12–14 kDa) in deionized water (DIW) for five days to remove unreacted MA and other by-products. The dialyzed solution was freeze-dried to obtain a white solid. The gelatin and GelMA in D_2_O (99.9 atom %D, Sigma-Aldrich) were measured using a Bruker Avance 600 MHz NMR spectrometer.

### 1.2. Preparation of Gelatin and ZnO-Loaded Gelatin Hydrogels

The ZnO-loaded gelatin hydrogel was prepared as follows. Firstly, GelMA (0.2 g) was dissolved in DI water (1.8 g) at 50 °C, and 2-hydroxy-4′-(2-hydroxyethoxy)-2-methylpropiophenone (20 mg, Sigma-Aldrich) was added as a photoinitiator. The solution was poured into a mold and exposed to LWUV light (Analytik Jena, Upland, CA, USA, 8W, irradiation distance 6 cm) at 365 nm for two hours. After curing, the resulting hydrogel was removed from the mold, obtaining a transparent film with a thickness of 2 mm. Hydrogel films were separately immersed in zinc nitrate solutions (J.T. Baker, Phillipsburg, NJ, USA) at different concentrations (1, 10, and 30 mM) overnight. After rinsing with DIW, the films were immersed in sodium hydroxide solutions (pH 10.6, Uniregion, Bologna, Italy) for two additional hours. Then, the samples were rinsed with DIW to obtain the Gelatin/ZnO1, Gelatin/ZnO10, and Gelatin/ZnO30 hydrogels. The hydrogels were subjected to freeze-drying for a duration of 24 h using a freeze dryer system, before being utilized for subsequent purposes.

### 1.3. Characterization of the Hydrogels

The transmission and absorption spectra of the samples were measured using a UV–vis spectrophotometer (TECAN infinite M200) with a scanning wavelength range of 250–800 nm. The characterization of ZnONPs, the gelatin hydrogel, and the composite hydrogels was carried out using a Nicolet Nexus 670 FT-IR spectrometer (Thermo Fisher Scientific, Waltham, MA, USA). FT-IR spectra were obtained by scanning the specimens 32 times in the wavenumber range from 500 to 3500 cm^−1^ at a resolution of 1 cm^−1^ with an attenuated total reflection (ATR) module at room temperature. The morphology and distribution of ZnONPs in the hydrogel were characterized using an ultra-high resolution scanning electron microscope (HR-SEM, Hitachi SU8010, Tokyo, Japan). The freeze-dried hydrogel samples were sputter-coated with Pt before analysis. Additionally, energy-dispersive spectrometry (EDS) measurements were performed on cross sections of the freeze-dried samples. The particle size of ZnONPs residing in the hydrogels was characterized by transmission electron microscopy (TEM, Hitachi H7500 microscope). The gelatin was degraded by bromelain (≥3 units/mg protein, Sigma-Aldrich). Specifically, 20 mg of the freeze-dried hydrogel was immersed in 1.0 mL of DIW with 3 mg of bromelain for 24 h, and the resulting solution was centrifuged to collect the ZnONPs. To quantify the content of zinc oxide in the hydrogel, 25 mg of the freeze-dried hydrogel samples was digested in 3 mL of nitric acid (Sigma-Aldrich) and heated at 70 °C for one hour. The measurements were conducted by using an inductively coupled plasma mass spectrometer (ICP-MS, THERMO-ELEMENT XR). The thermal properties of the hydrogel samples were analyzed using thermogravimetric analysis (TGA, Perkin Elmer TGA4000, Walthman, MA, USA) at a ramping rate of 10 °C/min in a nitrogen atmosphere. Dynamic mechanical analysis (DMA) was performed using a strain-controlled rheometer (ARES-G2, TA Instruments Japan Inc., Tokyo, Japan) to analyze the stress–strain behavior of the hydrogels. The samples were dispensed in aliquots with a diameter of 10 mm and a height of 5 mm using parallel plate geometry and a compression rate of 0.01 mm/s at room temperature.

### 1.4. Swelling Behavior of the Hydrogels

The swelling behavior of the hydrogels was investigated as follows. Well-prepared hydrogels (200 μL) were freeze-dried and then incubated in DTW (pH 6.8) at 37 °C for 12 h. After removing the excess water on the gel surface with tissue paper, the swollen hydrogels were weighed. The swelling ratio (SR) was determined using the weights of the swollen (*W_s_*) and freeze-dried (*W_fd_*) hydrogel samples according to the following formula:$$SR=\frac{W_{s}-W_{fd}}{W_{fd}}$$

### 1.5. Antibacterial Activity

The in vitro antibacterial activity of the ZnO-loaded gelatin hydrogels was investigated against Gram-negative bacteria, including enterohaemorrhagic *E. coli* (*EHEC*), *Shigella flexneri* (*S. flexneri*), and *Staphylococcus aureus* (*S. aureus*). The test bacteria were provided by the Bioresource Collection and Research Center (Hsinchu, Taiwan). Disc-shaped hydrogels (100 μL, 6.0 mm of diameter) were incubated with bacteria solutions (10^5^ colony-forming units—CFU mL^−1^) at 37 °C for 2 h. After incubation, 100 μL of the solution was plated on a Luria–Bertani agar plate (Cyrusbioscience) and incubated at 37 °C for 24 h. The number of bacterial colonies was counted to evaluate the percentage of bacterial cell reduction (R%) on the gelatin hydrogel and ZnO-loaded gelatin hydrogels using the equation reported in previous research [26]. *CFU_control_* is the number of CFU per milliliter in samples without the hydrogels. *CFU_sample_* is the number of CFU per milliliter in samples treated with a hydrogel.
$$R\%=\frac{{CFU}_{control}-{CFU}_{sample}}{{CFU}_{control}}\times 100\%$$

The images of live and dead bacterial cells were obtained through SEM. The samples containing the bacterial cells were prepared by incubating *EHEC*, *S. flexneri*, and *S. aureus* (10^7^ CFU mL^−1^) with and without Gelatin/ZnO30. To this aim, 1 mL of bacterial solution was centrifuged at 8000 rpm for 5 min, followed by washing the bacterial cells with 500 µL of 0.1 M PBS; this procedure was repeated 2–3 times. Next, 500 µL of solution A (50 µL of 25% glutaraldehyde, 250 µL of 0.2 M PBS, and 200 µL of water) was added to the cells, and the mixture was allowed to stand at 4 °C for 1.5 h. The cells were then washed again 2–3 times with 500 μL of 0.1 M PBS and resuspended in 800 μL of PBS. Meanwhile, 5 μL of Formvar solution (Sigma-Aldrich) was evenly spread onto silicon wafers and left to dry for 20 min. Then, 10 μL of the bacterial solution was dropped onto the silicon wafers, evacuated using a vacuum pump, and let dry for 30 min. A suitable amount of solution A was dropped onto the wafers and frozen for 20–30 min. The cells were washed again with 800 µL of 0.1 M PBS 2–3 times and then soaked in the buffer for 10 min. At this point, the cells were sequentially dehydrated by soaking in 30%, 50%, 70%, and 100% ethanol solutions for 7 min each. Subsequently, 500 µL of tert-butanol (J.T. Baker) was added to the silicon wafers, which were placed in a 4 °C refrigerator until complete tert-butanol crystallization. Finally, the silicon wafers were freeze-dried until complete evaporation of tert-butanol was achieved.

### 1.6. Cell Viability Assay

The in vitro biocompatibility of the hydrogels with NIH 3T3 and BEAS-2B cells was assessed according to a modified ISO 100993-5 standard method [27]. Hydrogel discs were prepared at a concentration of 0.1 g/mL in culture medium. Subsequently, the samples underwent ultrasound treatment and were washed twice with PBS to purify them and remove any residual material from the hydrogel network. The cells were then seeded onto a 96-well plate, with each well containing 5 × 10^3^ cells, and cultured in an essential medium consisting of a 10% CCS (Cosmic Calf Serum, HyClone) or FBS (Fetal Bovine Serum, HyClone) combined with 90% DMEM (Dulbecco’s Modified Eagle Medium, GeneDireX) for 24 h at 37 °C. After the incubation, the medium was removed, and 100 μL of 10% CCS or FBS medium was added to the wells. The purified hydrogels were introduced into the cell culture media and incubated for 24 h at 37 °C and 5% CO_2_. Following the incubation period, the hydrogel samples were removed, and a CCK8 solution was added to the wells, followed by a 30 min incubation. Cell viability was determined using the cell counting kit-8 (CCK-8), and the optical density (OD) was measured at 450 nm using an ELISA plate reader (Sunrise, Tecan).

## 2. Results

### 2.1. Characterizations of the GelMA Polypeptide and ZnO-Loaded Gelatin Hydrogel

The GelMA polypeptide was synthesized using gelatin and methacrylic anhydride in phosphate buffer (pH = 7.4) at 50 °C, as previously reported [28]. Methacryloyl (MA) substitution groups were introduced on the reactive amine and hydroxyl groups of the gelatin side chains (Figure 1).

Appendix A illustrates the ^1^H NMR spectra of gelatin and GelMA, with the spectrum of the GelMA polypeptide displaying the methacrylate groups at 5.7 and 5.8 ppm (a, b, =CH_2_) and the primary amines of gelatin at 3 ppm (ε, -CH_2_NH_2_), confirming the successful conjugation of MA to gelatin. The chemical shift at 1.9 ppm was attributed to the methyl groups of MA (d, -CH_3_). Moreover, the degree of substitution (DS) of gelatin by the methacryloyl groups was calculated by determining the ratio of the integration area of the methylene peaks from the chemical shift of gelatin and GelMA at 2.9 ppm in ^1^H NMR analysis and found to be 95.8%, which is consistent with the results reported in previous studies. A higher molar MA/gelatin ratio corresponded to a higher degree of substitution of gelatin [29].

### 2.2. Characterization of the Gelatin and ZnO-Loaded Gelatin Hydrogels

Upon the in situ formation of ZnONPs in the gel network, the appearance of the composite hydrogel samples exhibited no difference compared to their unloaded counterpart (Figure 1a). The characterization of the ZnO-loaded gelatin hydrogels was carried out using FTIR spectroscopy. The IR spectrum of the gelatin hydrogel showed a broad band at 3334 cm^−1^ and a band at 1632 cm^−1^ (Figure 1b), attributed to the tensile vibration of the N-H bond and the vibration of the C=O bond of amide I, respectively [30]. ZnONPs were prepared by the precipitation via adding NaOH to a Zn(NO_3_)_2_ solution. It can be seen that the characteristic bands of both gelatin and ZnONPs were present in the IR spectrum of Gelatin/ZnO30, indicating the successful in situ formation of ZnONPs in the gelatin hydrogel.

A quantitative analysis of ZnONPs in the composite hydrogels was conducted using ICP-MS analysis. As shown in Table 1, the zinc content in the as-prepared hydrogels, corresponding to the content of ZnONPs, increased with the increase in the concentration of added zinc nitrate. Previous studies reported that the composite hydrogels exhibited in situ formed ZnONPs with particle sizes larger than 30 nm [19,31]. The particle size distributions and Gaussian fittings of the gelatin/ZnO composite hydrogels are shown in Appendix A. As summarized in Table 1, the average mean particle size of the gelatin/ZnO composite hydrogels was much smaller than 30 nm, significantly smaller than those previously reported. Moreover, there was no significant difference between the three samples, indicating that the gelatin hydrogel networks could prevent the agglomeration of the NPs upon their in situ formation. The transmittance of the hydrogels between 280–800 nm was analyzed using a UV–vis spectrophotometer. The transmittance at 280–300 nm of all hydrogel samples decreased rapidly (Figure 1c) [32]. It can be observed that the transmittance at 300–800 nm decreased with the increase in ZnONP content.

The microstructure of gelatin hydrogel sections was found to present pores (Figure 2). EDS mapping was used to study the element distributions in Gelatin/ZnO30 cross-sections, and the mapping result (Figure 2b) confirmed that the ZnONPs were embedded inside the gelatin hydrogel and were uniformly distributed from the surface to a specific depth. There was no obvious aggregation of zinc oxide particles between the sections of the Gelatin/ZnO30 hydrogel (Figure 2a), indicating that the size of the particles was in the nanoscale. In addition, comparing the pore sizes of gelatin, Gelatin/ZnO1, Gelatin/ZnO10, and Gelatin/ZnO30 (Figure 2a, Appendix A), we observed that as the concentration of Zn^2+^ increased, the network of the hydrogels became more tightly connected, the pores in the hydrogels became smaller, and the composite hydrogels exhibited fewer pores than the gelatin hydrogel. These results are different from those obtained when doping metal nanoparticles into hydrogels, as previous studies indicated that metal nanoparticles reduced the wall thickness of hydrogels [33] and increased the pore size, due to the accumulation and agglomeration of the nanoparticles [34]. This is consistent with the results of subsequent swelling tests and mechanical property measurements.

The TGA method was used to compare the thermal behavior and thermal stability of different hydrogels. The thermal decomposition process (Table 1 and Appendix A) was divided into two stages [35]. In the first stage (50~165 °C), the weight loss was about 5%, which was caused by the removal of absorbed water. The second stage was attributed to the decomposition of the hydrogel. The weight loss of gelatin and the ZnO-loaded gelatin hydrogels ultimately decreased with the increase in the content of ZnONPs at 750 °C. Compared to gelatin, the ZnO-loaded gelatin hydrogels showed a significant improvement in thermal stability, with residual mass beyond the expectations; similar results were obtained in a previous study [36]. The anomalous thermal decomposition behavior can be credited to the chelation effect of the NPs that increased the stability of the gelatin hydrogels. As a result, the ZnO-loaded gelatin hydrogels had higher thermal stability and lower weight loss compared with the gelatin hydrogel without ZnONPs. At 300 °C, there was only a 10% weight loss, indicating that these hydrogels are thermally stable within the temperature range of usual medical applications.

The mechanical properties of the gelatin hydrogel and ZnO-loaded gelatin hydrogels were tested using a strain-controlled rheometer. The stress–strain curves of hydrogels with different Zn^2+^ concentrations are shown in Table 1 and Appendix A. The ultimate stress and ultimate strain increased as the zinc concentration increased, suggesting that the presence of the ZnONPs contributed to enhancing the mechanical strength of the hydrogel. Young’s modulus was calculated in the elastic region of the hydrogels, i.e., the region with a strain of 0–15%. The curves overlapped in the elastic region, indicating that the different loading contents of ZnONPs did not affect Young’s modulus, which was 3.81 × 10^−4^ MPa for all hydrogels. A previous study by Bikendra Maharjan also demonstrated similar GelMA stress–strain curves and showed that the compressive strength or stress-bearing capacity increased with the addition of Au and SiO_2_ NPs [37], in accordance with the amount of added NPs. However, Young’s modulus variation was not clearly associated with the presence of Au and SiO_2_ NPs.

### 2.3. Swelling Behavior of the Hydrogels

The appropriate swelling ratio of a hydrogel ensures its stability and the absorption of wound exudate. The swelling behavior of the Gelatin/ZnO1, Gelatin/ZnO10, Gelatin/ZnO30, and gelatin hydrogels in DTW is shown in Figure 3. With the increase in ZnONPs, the swelling ratio slightly decreased. this could be attributed to the formation of hydrogen bond interactions between the ZnONPs and gelatin, leading to the increase in the cross-linking density of the hydrogel structure and the improvement of its mechanical properties.

### 2.4. Antibacterial Activity of the Hydrogels

The antibacterial properties of the gelatin hydrogel and the ZnO-loaded gelatin hydrogels were evaluated against Gram-negative and Gram-positive bacteria using the CFU assays. The results revealed that the survival of the bacteria decreased with the increase in ZnONPs content (Table 2 and Appendix A). One previous study indicated that the MIC values of a composite hydrogel of poly(vinyl alcohol), chitosan, and ZnONPs against *S. aureus* and *E. coli* were 50 μg mL^−1^ and 200 μg mL^−1^, respectively [34]. Another study reported that the MIC_50_ value of ZnONPs against *S. flexneri* was 42 μg mL^−1^ [38]. The particle sizes of ZnONPs in these two studies were much greater than 20 nm. In this study, the ZnO-loaded gelatin hydrogels demonstrated a bacteriostatic capability, with an over 70% reduction in bacterial cell count upon exposure to ZnONP concentrations of 8 µg mL^−1^. Additionally, they exhibited nearly complete bactericidal activity at ZnONP concentrations exceeding 16 µg mL^−1^. Our study demonstrated that the in situ formation of ZnONPs allowed for much smaller particle sizes and better dispersion than other methods reported in previous studies, resulting in larger surface areas and, consequently, better antibacterial efficacy of the ZnO-loaded gelatin hydrogels. The ZnONPs exhibited a stronger antibacterial effect against Gram-positive bacteria than against Gram-negative bacteria, which is consistent with previous studies [39,40]. The higher sensitivity of Gram-positive bacteria to ZnONPs may be related to differences in cell wall structure, cellular physiology, and metabolic processes with respect to Gram-negative bacteria [41]. In addition, *S. aureus* secretes gelatinase, which leads to the degradation of the gelatin hydrogel network, resulting in a superior antibacterial activity of the hydrogels compared to that observed against *EHEC* and *S. flexneri* [42]. Moreover, previous studies determined the antibacterial properties of ZnO composite hydrogels by employing the inhibition zone method [19,21]. These investigations consistently demonstrated antibacterial efficacy within a defined concentration range, thereby implying the release of ZnONPs from the hydrogel. This phenomenon suggests that the antibacterial activity extends beyond the hydrogel surface.

The SEM images confirmed the morphological changes caused by ZnONPs in the bacteria. The Figure displays images of *S. flexneri* (a, d), *EHEC* (b, e), and *S. aureus* (c, f) in the absence (a–c) and presence (d–f) of the Gelatin/ZnO30 hydrogel (Figure 4). There is a noticeable difference in morphology between bacteria treated or not with the ZnO-loaded gelatin hydrogels. The untreated bacteria exhibited an intact cell structure with rod-shaped or spherical morphology. The bacteria subjected by the bactericidal action of the ZnONPs displayed irregular cell surfaces due to cell lysis, resulting in ill-defined morphology and boundaries [43]. The primary mechanism of the bactericidal action was attributed to lipid peroxidation induced by reactive oxygen species generated by the ZnONPs, leading to the structural damage in the cells [44,45].

### 2.5. Cell Viability Assay

Previous studies showed that gelatin hydrogels and ZnONPs exhibited excellent cellular compatibility [46,47,48]. Figure 5 shows the viability of NIH 3T3 and BEAS-2B cells upon exposure to gelatin and the ZnO-loaded gelatin hydrogels. Upon treating with the gelatin hydrogel, the viability of NIH 3T3 and BEAS-2B cells was close to 100%, consistent with the results of previous studies [49,50]. The introduction of ZnONPs led to a slight decrease in cell viability, which could be possibly attributed to the release of toxic Zn ions from ZnONP dissolution. All the hydrogels exhibited excellent cellular compatibility, evidenced by cell viability values higher than 70%.

## 3. Conclusions

In summary, the ZnO-loaded gelatin hydrogels were prepared via the in situ formation of ZnONPs in a gelatin hydrogel and displayed remarkable biocompatibility and antibacterial activity, mainly attributed to the formation of nano-sized (12–14 nm), well-distributed ZnONPs in the hydrogel network. In comparison to traditional synthetic approaches, in situ formation has environmentally friendly characteristics, low toxicity, and excellent properties in general. As the concentration of ZnONPs increased, the ZnO-loaded gelatin hydrogels displayed enhanced mechanical properties and thermal stability, reduced swelling ratio and transmission, and better antibacterial activity. The hydrogels exhibited antimicrobial effects against both Gram-negative and Gram-positive bacteria, with a more pronounced antibacterial effect against Gram-positive bacteria. Hydrogels with a ZnO concentration over 0.16 μg [Zn]/mg [Hydrogel] could kill over 99% of the tested bacteria. These findings suggest that ZnONPs composite hydrogels hold great potential for future applications in the field of biomedical engineering.

## Data Availability

The data presented in this study are available on request from the corresponding author.

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
