# Peer review of "Antibacterial Gelatin Composite Hydrogels Comprised of In Situ Formed Zinc Oxide Nanoparticles"

_polymers, 2023, doi:10.3390/polym15193978_

Round 1

Reviewer 1 Report

Yu Y.-C. and co-workers reports about the in-situ nucleation of zinc oxide nanoparticles inside gelatin hydrogel. The quality of the work is good and I agree with publication of the manuscript after minor corrections reported below:

Figure 1a: please check the label of x-axis.

Table 1 and Table 2: adjust the standard error value according to the number of significant digits.

Figure 2b: the error bar is missing.

Did the author evaluate the quantity of Zn2+ released by the hydrogel? Some quantitative release studies should be performed.

The quality of the english is overall good. Please check some mistakes and typos through the text.

Reviewer 2 Report

This is an interesting study about antibacterial gelatin composite hydrogels comprised of in-situ formed zinc oxide nanoparticles. I suggest it for publication after the following minor points are addressed.

1. Line 43-45, one relevant study (Science 370 (6514), 335-338) should be included to support such a claim. 

2. In Scheme 1, why were only amine and hydroxy groups drawn for the gelatin molecule?

3. The resolution of Figure 1 should be improved.

4. The cytotoxicity of ZnO-loaded gelatin hydrogels is not great (cell viability less than 90% after incubation for 24 h). The authors should comment on this point.

5. Are these ZnO released from the hydrogel during the application?

Minor editing of English language required

Reviewer 3 Report

The paper submitted by Yu et al. deals with the preparation of a series of hybrid hydrogels loaded with ZnO nanoparticles which were obtained in-situ. The authors demonstrated the high antimicrobial activity of these hydrogels.

The manuscript is clear, well written and the conclusions are supported by the results. However, some corrections are needed:

1. the introduction section must be completed with the following recent reference: https://doi.org/10.3390/pharmaceutics15092240

2. lines 75-77: this sentence must be deleted as not results can be given in the introduction.

3. materials section is missing.

4. how the hydrogels were dried?

5. delete lines 283-286.

6. As expected the hydrogels with 30% ZnO are becoming to be cytotoxic after 24h and I suppose that the cell viability is much lower after 48h. The authors must discuss also about that in more detail.

7. more specific results should be provided in the conclusion section.
